# Impairment of visual and neurologic functions associated with agrochemical use

**Ingrid Astrid Jiménez Barbosa** [1]*, **Martha Fabiola Rodríguez Alvarez**[2], **Laila Cristina Bernal Bechara**[3], **Sieu K. Khuu**[1]

1 School of Optometry and Vision Science, The University of New South Wales, Sydney NSW, Australia,
2 Health Sciences Faculty, Program of Optometry, Universidad de La Salle, Bogotá, Colombia, 3 Agricultural Sciences Faculty, Program of Animal Science, Universidad de La Salle, Bogotá, Colombia

☯ These authors contributed equally to this work.
* ingrid.jimenezbarbosa@unsw.edu.au

**Data Availability Statement:** All relevant data are within the paper and its Supporting Information files.

## Abstract

To determine whether exposure to occupational levels of agrochemicals is associated with a range of low- (contrast and colour) and higher-level visual functions, particularly the detection of global form and motion coherence. We compared the performance of workers exposed to occupational levels of pesticides and non-exposed individuals on visual tasks that measured colour discrimination (Farnsworth Munsell 100 and Lanthony D15 desaturated) and the contrast sensitivity function (1–16 cpd). Global form and motion detection thresholds were measured using Glass-pattern and global dot motion stimuli. Neurotoxicity symptoms and biological markers associated with pesticide exposure were quantified using the Q16 modified questionnaire and via tests for levels of acetylcholinesterase in blood and substance P from the tear film, respectively. Workers exposed to pesticides had significantly more neurotoxic symptoms than non-exposed workers. No significant difference between groups for acetylcholinesterase levels was found, but there was a significant group difference in Substance P. The exposed group also had significantly poorer contrast sensitivity, colour discrimination and higher coherence detection thresholds for global form and motion perception. Exposure to occupational levels of agrochemicals in workers with signs of neurotoxicity is associated with low and high visual perception deficits.

## Introduction

Modern agricultural production systems have conventionally used agrochemicals, including fertilizers and pesticides [1]. The most commonly used agrochemicals are pesticides such as organophosphates (OPs), carbamates (CAs), pyrethroids (PIs), and neonicotinoids (NEs), which act to block key processes in the central nervous system of insects [2]. OPs and CAs act by inhibiting acetylcholinesterase (AChE), which is a catalytic enzyme involved in the breakdown of acetylcholine (ACh) to acetate and choline in the synaptic clefts of neurons [3]. This action disrupts normal CNS function, and the accumulation of ACh leads to peripheral signs such as increased sweating and salivation, lacrimation, miosis, blurred vision, tachycardia,

**Funding:** This study was funded by Universidad de La Salle, Bogota, Colombia, and The University of New South Wales under Grant # VRIT_CISVI 19072017 to IAJB. The funders had no role in study design, data collection and analysis, decision to publish or preparation of the manuscript.

**Competing interests:** The authors have declared that no competing interests exist.

respiratory paralysis, increased gastrointestinal motility and tremors and can result in severe neurotoxicity and cognitive impairments [4].

The plasma and erythrocyte AChE activity in human workers can be measured as an indicator of the accumulation of OPs and CAs in the body. This method is commonly used to measure pesticide exposure or poisoning [5] and to monitor workers who may be at an increased risk of exposure to OPs, such as those who work in agricultural and chemical industries. This test is an important biomarker of nervous system change following exposure to OPs and CAs in occupational and clinical environments [6].

In recent years, several studies have begun to recognize the deleterious effect of pesticides on brain function, particularly on vision, the visual system and ocular tissue, where the chemical injury can be high [7–9]. The sensation of irritation, burning, stinging or pain in the eyes is a cognitive and emotional process signalled at somatosensory terminals of the trigeminal nerve, mainly involving Substance P (SP) and the calcitonin gene-related peptide (CGRP), which are pro-inflammatory substances [10]. Additionally, SP is a neuropeptide released by striatal neurons, which play a critical role in regulating motor (including eye movements) and cognitive functions [11].

The most common visual symptoms reported by agricultural workers exposed to pesticides are changes in near vision, far vision, blurred vision, watering of the eyes, hyperemia, pain in the eyes, and swollen eyes among pesticide sprayers [12]. It has been reported that organic pesticide exposure can also result in eye-related disorders such as pupilar changes, accommodative and refractive variations, optic disc edema, and optic nerve atrophy [13].

Initial reports have indicated that vision can be affected by exposure to pesticides. However, it remains unclear how pesticide exposure may affect other aspects of the visual system, particularly visual function and perception, which are primarily a product of neural processes in visual areas of the brain. The goal of the present study adopted a similar approach to Jiménez et al., [14] and we evaluated the extent to which both low- and higher-level visual function is associated with exposure to agrochemicals and, in particular, pesticides.

## Materials and methods

### Study population

This was a cross-sectional case control study that included fifty workers (the exposed group) from fourteen agricultural systems/farms, and 50 normal participants (non-exposed /control group) was recruited to participate in the present study. This sample size (conforming to a power of 80.0% and Type I error probability of 5.0%) was determined using the PASS software based on the means for exposed to organic solvents (48.0 SD±1.31) (and for non-exposed 21.65 SD±1.65) on the modified Q16 questionnaire from a previous study [14–16].

The inclusion criteria for cases were workers exposed to agrochemicals for at least 3 years and with at least medium-low level (score: 33–48) of neurotoxic symptoms as indicated by the modified Q16 questionnaire. Participants comprising the exposed group were aged between 18 to 45 years. For the control group, participants were not exposed regularly to any agrochemical substance. This was established in an interview and by the modified Q16 questionnaire in which the level of neurotoxicity was less than 32 (i.e., less than low level). Control participants had the same educational level as exposed workers and lived in the same geographic location at an approximate distance of 15 km from the agricultural farms.

The exclusion criteria for controls and cases were participants who, at the time of visual assessment, were those who had systemic infections, heart, and liver cancer disease, congenital colour vision deficiencies (assessed with Ishihara test), systemic and neurological disorders unrelated to environmental toxins or women who were pregnant.

Agricultural workers at the flower farms used agrochemicals such as OPs: Lorsban (Chlor-pyrifos) (O, O-Diethyl O-(3,5,6-trichloropyridin-2-yl) phosphorothioate) and Monitor Profi-col (Methamidophos) (O,S-Dimethyl phosphoramidothioate). PIs: Tordon (Picloram, herbicide) (4-Amino-3,5,6-trichloro-2-pyridinecarboxylic acid); fungicides such as Citrolife, and CAs: Antracol (Zinc polimérico 1,2 propilenbis (ditiocarbamato). The use of pesticides was associated with three basic activities/tasks, 1) mixing and loading the agrochemical prod-uct, 2) application of the spray solution, and 3) cleaning of the spraying equipment. Though using protective equipment (e.g., face masks, gloves, and protective clothes) in handling pesti-cides is a requirement by law, adherence to regulation is poor and not regularly followed, nor were regular logs regarding the frequency of use of protective equipment kept. Workers may rotate between these three tasks. Typically, farms utilise approximately 1.64 kg/ha of agro-chemicals and pesticides annually. Over four weeks, the type and frequency of use of pesticides by applicators and operators collaborating on farms who participated in the present study were: Ops: 22 times; Pis: 16 times; Cas: 22 times.

Each participant completed the following forms: a demographic information questionnaire and a baseline survey (used to estimate the exposure to pesticides). A Log Mar chart was used to assess distance visual acuity, and a reading chart was used to evaluate near visual acuity (40 cm). Biomicroscopy (Topcon) was used to assess the presence or absence of lens opacities using the lens opacities classification system III (LOCS III) [17]. Direct ophthalmoscopy and digital retinography (DRS Centervue) were used to assess ocular health. The participants were refracted using an autorefractor (Unicos URKR-7000), and those who needed refractive cor-rection were corrected with untinted lenses. Colour vision was evaluated for red-green con-genital colour vision deficiency using the Ishihara test (24 plates). For the exposed group, all visual tests, surveys and the taking of biological samples were conducted on-site on all on the same day and during their work shift in a dedicated testing room. All study participants pro-vided written consent in accordance with the Declaration of Helsinki, approved by the ethics committee of the Faculty of Health Sciences of the University of La Salle.

## Measures of neurotoxicity exposure and levels of substance P in tears

**The modified Q16 neurotoxic symptoms questionnaire.** Q16 questionnaire was vali-dated in Spanish and is commonly used to monitor for neurotoxic symptoms in working pop-ulations [18]. The modified version of the questionnaire has 16 questions regarding neurotoxic symptoms, and the answers were graded on a Likert scale with five levels of response (stronger disagree; disagree; neutral; agree and strongly agree). The questionnaire was administered as an interview by the investigators to ensure a good understanding of the questions and provided an opportunity for the participant to ask and clarify any doubts they may have. The modified version can be scaled as 6 to 32 Low; 33 to 48 Medium-Low; 49 to 64 Medium; 65 to 80 High [15,16].

**Test for cholinesterase (AChE) in blood.** A sample was taken with EDTA anticoagulant by a professional bacteriologist, and the plasma was separated by centrifugation (1200 rpm x 10 min) and transported to the laboratory in an ice chamber within 6 hours. The Wiener lab® brand kit and the fixed ΔT procedure were used. The reference values are 3200–9000 U/I.

**Tear Substance P (SP).** A tear sample was taken from the conjunctival sac of the selected eye with a sterile swab, which was transported to the laboratory in an Eppendorf tube with 200 microliters of sterile saline solution in an ice chamber. Samples were stored at -20˚C. The commercial kit CAYMAN chemical ® was used, which uses the ELISA tech-nique by competition. The results were read in the Mindray ELISA microreader at 405 nm

and reported in pg/ml. The reference values for the concentration of SP in human tears are between 306.0 +/- 96.5 pg/ml [19]

All visual tests were conducted in a dimly lit room using a 21-inch Macintosh laptop using custom software written in MATLAB. Stimuli were presented on the laptop screen, and participants viewed the stimulus at 70cm. All tests were conducted monocularly using the eye, with the worst best corrected visual acuity (VA) being 0.1 Log MAR.

## Low-level visual function tests

**Contrast sensitivity.**   The contrast sensitivity function (CSF) was assessed through a computerized test using custom-written software in MATLAB (version 11). Participants were presented with an oriented Gabor patch (diameter of 4 ˚) and had to judge whether the pattern was tilted 45 ˚ to the right or left of vertical in a two-alternative forced choice design (Fig 1). The background was mid-grey and at a luminance of 55 cd/m$^2$. The Weber contrast of the stimulus, coinciding with the amplitude of the Gabor stimulus, was modified using a staircase procedure corresponding to the 79.0% correct performance level. Initially, the starting contrast of the stimulus was 0.8, and the step size was 0.08. After the first and subsequent reversals, the step size was halved. After the third reversal, the step size was 0.01 and remained at this value until the end of the staircase trial. The staircase lasted six reversals, and the average of the last four reversals was averaged to estimate the contrast detection threshold. No feedback was given to indicate the correctness of the response. The test was conducted monocularly, and the stimulus presentation was 1 second. The staircase procedure was repeated for the following spatial frequencies: 1.0; 2.0; 4.0; 8.0, and 16.0 cycles per degree (cpd), and each observer was measured once at each spatial frequency in a randomized order.

**Farnsworth-Munsell 100-Hue test.**   The test consists of 85 movable colour caps arranged in four boxes of 21 or 22 colours each. The colours are set in plastic caps and subtend 1.5 ˚ at 50 ˚ and numbered according to their correct colour order in a circle. One box is presented at a time. The observer had to arrange the caps according to colour in a lower tray with two caps as references. The FM Hue 100 test was performed to obtain a complete analysis of the colour vision deficiencies.

**Lanthony D15 (desaturated) test.**   The Lanthony D15d is a desaturated colour vision test that measures fine colour discrimination and has been used widely in analyzing acquired colour vision deficits [20] The test consists of 15 coloured caps (subtending a visual angle of 1.5˚ at 50 cm) located in a box with one reference cap at a fixed location. The colours are much paler and lighter than and appear almost white. The Lanthony Desaturated D-15 test has been used to document colour discrimination ability after occupational exposure to organic solvents such as styrene and toluene [21].

## High-level visual functions tests

**Global dot motion.**   Circular Global Dot Motion stimulus (diameter 5˚ of visual angle) comprised of 500 moving light-increment dots (40 cd/m$^2$ radius 0.06), randomly distributed on a grey background (30 cd/m$^2$) (Fig 1B) presented for 500 ms. All dots moved at a speed of 3 deg/s, and a proportion of dots moved coherently (signal) amongst the remaining randomly moving dots (noise). Signal dots move in a radial pattern direction as it has been shown that this type of complex motion is selectively processed in higher cortical areas [22].

Motion coherence thresholds were measured using a two-interval forced choice design in conjunction with a modified staircase procedure. In one interval, a global motion stimulus was presented containing signal dots and, therefore, radial motion for 500 ms. The global dot motion stimulus interval had only randomly moving dots. Both intervals were separated by a

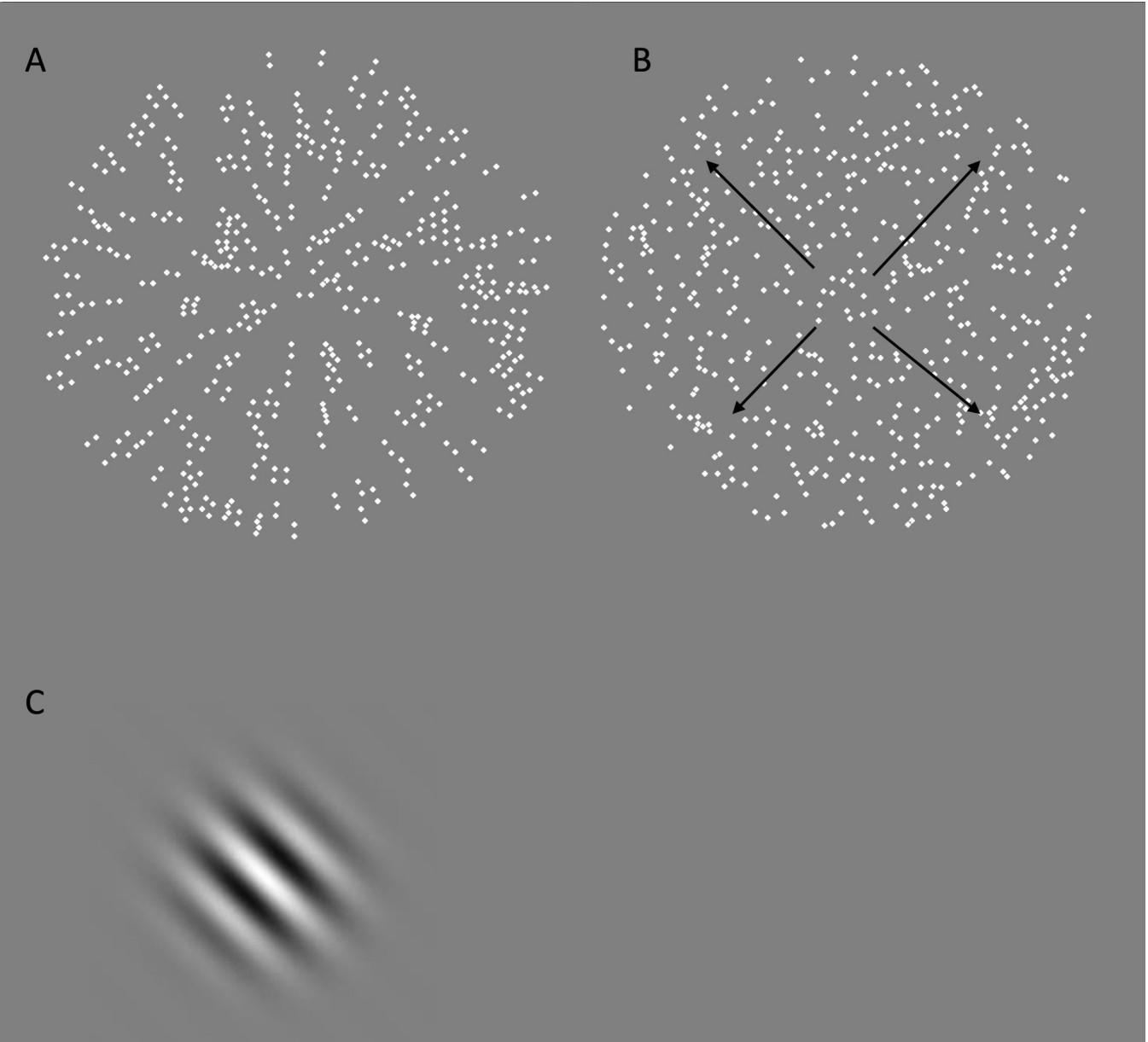

**Fig 1. Schematic examples of visual stimuli used to measure global form and motion.** (A – Glass pattern & B – global dot motion) and contrast sensitivity (C- Gabor patch).

period of 500 ms in which the screen was blank at the background luminance. The interval containing the coherent pattern was randomized from trial to trial. The participant's task was to judge the interval containing the stimulus with radial motion. A staircase procedure (corresponding to the 79.0% correct performance level) was used to change the number of signal dipoles in the image from trial to trial. The starting signal level of the staircase was 30.0% of the total number of dots, and the initial step size was 8.0%. The step size was halved on subsequent reversals until a step size of 1% of the total number of dots was reached (after the fourth reversal). The staircase remained at this step size until the end of the trial. The staircase lasted

for eight reversals, and the arithmetic mean of the last four reversals provided an estimate of the threshold.

*Global form perception* was measured using circular radial Glass patterns (diameter of 5˚) comprised of 250 dot-pairs (40 cd/m$^2$ radius, 0.06˚, dot-pair distance of 0.125˚) within a grey background luminance of 30 cd/m$^2$. A proportion of dot-pairs were signal and oriented in a radial pattern, while the remaining were randomly oriented (Fig 1A). Analogous to the methods used to measure global motion,

Glass pattern coherence detection thresholds were measured using a two-interval forced choice task in conjunction with a modified staircase procedure. In one interval, a coherent radial Glass pattern was presented for 500 ms, and the other interval contained a Glass pattern with only randomly oriented dot-pairs also shown for 500 ms. Both intervals were separated by an inter-interval period of 500 ms in which the screen was blank. The participant's task was to judge the interval containing the radial Glass pattern. A staircase procedure (corresponding to the 79.0% correct performance level) was used to change the number of signal dot pairs present in the image from trial to trial. Initially, the staircase presented 80 signal dot pairs, and the step size was eight dot pairs. The step size was halved after each reversal, and after the third reversal, the step size remained at one dot pair. The staircase lasted eight reversals, and the average of the last four reversals provided a measure of the threshold.

## Statistical analysis

Descriptive analyses included participant demographics, including age, gender and occupation were reported. A t-test or Mann-Whitney U test was used to determine statistical significance in the bivariate analyses. Multivariate analyses were performed using a two-way ANOVA. In all cases, a $p < 0.05$ value was considered statistically significant. The analysis was conducted with the R software, version 4.0.2 (The R Foundation for Statistical Computing).

## Results

Fifty exposed workers were included in the study. The average age was 35.2 years (SD±9.2) (range 21 to 45 years), and 82.0% were women. The levels of education of participants were high school (44.23%), primary school (38.46%), technical (15.38%) and bachelor's degree (1.92%). Sixty percent (60.0%) of workers were cultivation operators (i.e., they carry out cultural care, work in the greenhouse and were exposed to various temperatures, humidity, and solar radiation conditions, depending on the climate). 22.5% were room operators (these individuals worked indoors sorting flowers under artificial lighting and standard ventilation), and 17.5% were fumigators (they perform periodic fumigation using regulatory personal protection elements). All fumigators were male and did not always fulfil this function but also rotated for roles or performed transport functions. The average working time of the cultivation operators was 10.37 (SD± 6) years; room operators were 6.21 (SD± 4) years, and fumigators, on average, worked for 5.14 (SD± 7) years. The average working hours for all workers were 8 hours per day.

Fifty subjects with an average age of 30.69 years (SD±8.30 years) formed the control group, and 82.05% were female. The school level was distributed as follows high school (54.90%), technical (37.25%), technologist (5.88%), and bachelor's degree 1.96%. The total of the participants fulfilled the inclusion and exclusion criteria.

## Measures of neurotoxicity exposure and levels of substance P in tears

**Modified Q16 questionnaire.** Overall neurotoxicity score from the modified Q16 was compared between exposed and non-exposed groups. A Mann-Whitney U test confirmed that

the exposed group had significantly more neurotoxic symptoms than the non-exposed group (U = 515.0 p<0.0001). We find that the neurotoxicity level was medium-low at 40.51 (SD± 7.4) for the exposed group, while the non-exposed group had, on average, low levels of neuro-toxicity: 28.62 (SD±4.0).

**Tear Substance P.** The median level of SP for the exposed group was 225 pg/ml (RIC = 292.5 pg/ml), while in the non-exposed group, the median level was 96.5 pg/ml (RIC = 86.0 pg/ml). There was a statically significant difference between both groups (p<0.0006), showing that the exposed group had more SP in their tear film than the non-exposed group, and this suggests that a neuroinflammatory process could be present due to the exposure to pesticides.

**Cholinesterase analysis.** The median AChE level for the exposed group was 8732 U/I, while in the non-exposed group, the median level was 7369 U/I. There was no significant difference between the groups (p = 0.98). Two percent of subjects exposed to agrochemicals showed abnormal levels of cholinesterase below 3000 U/I, 11% presented levels between 3000 and 5000 U/I and 87% showed levels above 5000 U/I. In the non-exposed group, 100% of values were greater than 5000 U/I.

## Low-level visual function

**Contrast sensitivity.** Contrast sensitivity (1/Threshold) is plotted as a function of spatial frequency for both exposed and non-exposed groups in Fig 2. This figure shows that overall contrast sensitivity was significantly poorer in individuals exposed to pesticides than those not exposed. A two-way ANOVA showed a main effect of Group (F $(1.0,71)$ = 41.24, p<0.0001) and Spatial Frequency (F $(3.74, 265.8)$ = 13.17, p<0.0001), but no significant interaction effect F$(4.0, 284)$ = 0.03454, p = 0.997). These findings indicate that poorer performance detecting contrast for individuals exposed to pesticides was consistently lower than the non-exposed group across all spatial frequencies. Indeed, post hoc comparison tests (corrected for multiple comparisons) indicated significantly poorer contrast detection for the exposed group at all spatial frequencies.

**Colour vision tests.** A Mann-Whitney U test was performed to determine whether there were any differences between the exposed and non-exposed groups for FM Hue 100 TES and $\sqrt{}$TES. Both values were significantly higher in exposed than non-exposed workers (TES: U = 350, p≤0.0001) and ($\sqrt{}$TES U = 364, p≤0.0001), respectively. $\sqrt{}$TES was higher in exposed workers compared to non-exposed (Exposed workers: 11.43 (SD±3.63); non-exposed 6.50 (SD±.2.32)). $\sqrt{}$TES values for the non-exposed group were normal according to the group of ages established by Kinnear et al. [23]. Exposed workers showed higher total error scores than non-exposed participants, indicating poorer colour discrimination. Tritan-type errors in the exposed group were 61.0%, and diffuse discrimination was 39.0%. In the non-exposed group, Tritan-type errors were 4.0%, and diffuse discrimination was 16.0%.

**Lanthony D15d.** A Mann-Whitney U test was performed to determine whether the differences in TES (total error score) and colour confusion index (CCI) scores for the exposed group TES 64.41 (SD± 8.47); CCI 1.45 (SD ± 0.09) were significantly higher than for the non-exposed group TES 22 (SD± 6.20); CCI 1.16 (SD ± 0.04). TES: U = 746, p <0.0001) and (CCI: U = 762, p<0.0001). Tritan-type errors in the exposed group were 17.3%, Protan-type errors 1.9% and diffuse discrimination 13.4%. In the non-exposed group, 5.7% presented Tritan-type errors and 3.8% diffuse.

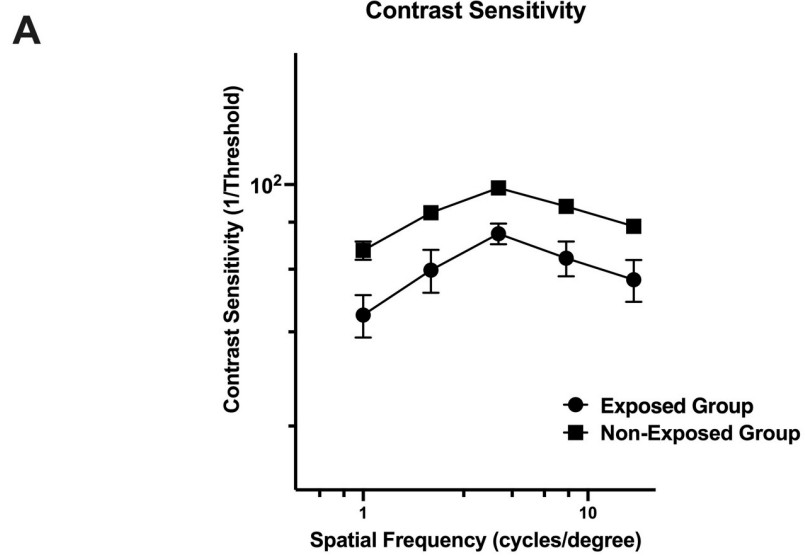

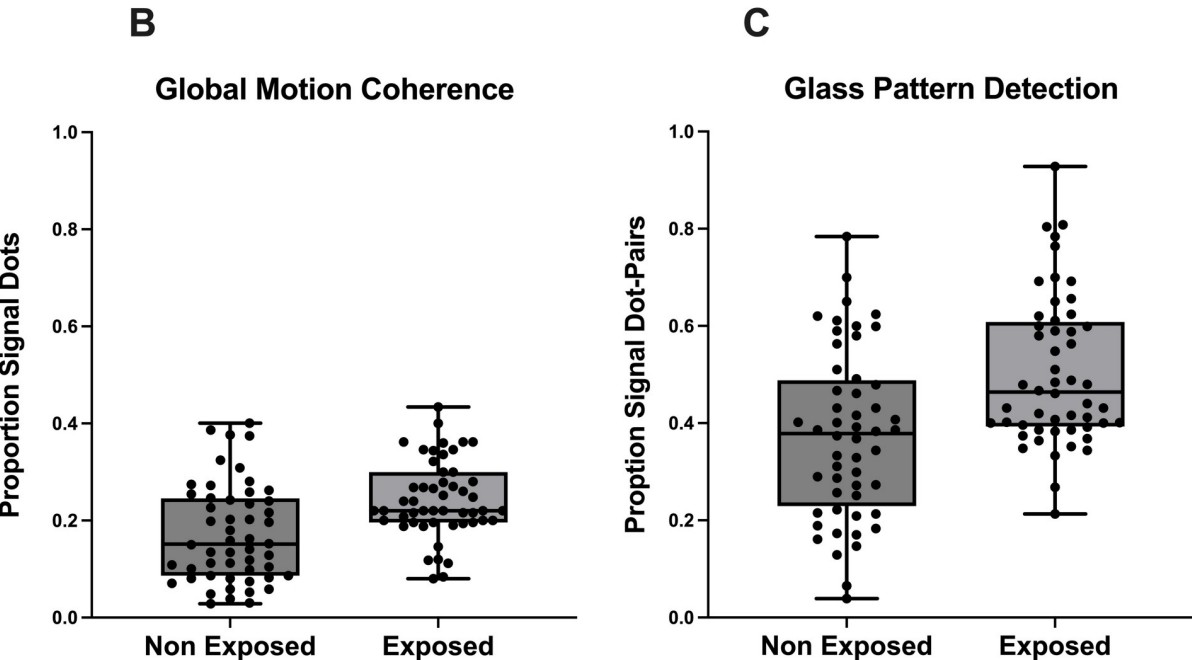

**Fig 2. Results for functional tests of vision and perception.** (A - contrast sensitivity, B - global motion and C- Glass pattern) for groups exposed or not exposed to pesticides.

### High-level visual function

We find that global form and motion coherence thresholds were higher in the exposed group than in the non-exposed group (Fig 2). The average global motion threshold was 11.0% for the non-exposed group, while for the exposed group, the threshold was 17.0%, which was significant (U = 768.5, p = 0.0002). A similar pattern of results was reported for global form

perception. The average global form coherence threshold was significantly higher (U = 785, p = 0.0002) for the exposed group (36.0%) than for the non-exposed group (50.0%).

## Discussion

The *Guidelines on highly hazardous pesticides* recognize that agrochemicals may deleteriously affect human, animal, and environmental health. This study confirms this observation by demonstrating an association between changes in visual function in individuals exposed to occupational levels of pesticides.

In the present study, individuals exposed to pesticides had significantly higher Q16 symptom scores and higher levels of SP, which is likely to be associated with their exposure to pesticides and the impact of pesticides on nervous system functioning. These findings are consistent with previous studies [14, 24], which have shown that exposure to occupational levels of organic solvent can lead to neurotoxic symptoms. However, there was no significant difference in AChE levels between individuals in exposed and non-exposed groups. A possible reason is that the farm workers included in the present study were regularly monitored for AChE, as per the Panamerican Health Organization (PAHO) guidelines. Here following regulations may limit the exposure of pesticides to a point where it is minimally present in blood. Note, however, that AChE levels are likely to be variable and have been dependent on other factors such as weight, health and the consumption of alcohol or smoking [25] which were not controlled for in the present study. Future studies may examine whether and how these factors influence the measurement of AChE levels in individuals exposed to agrochemicals.

Whilst no difference in AChE levels was reported between the two groups, a significant increase in SP levels was observed in workers exposed to agrochemicals. After corneal injury by chemicals, SP and its receptor NK1 concentration rapidly increase in cornea-induced neurogenic inflammation in wound healing, inflammation, and pain [26]. The current research is the first to measure SP concentration in tears as an indicator of ocular surface damage from pesticide exposure. The increase in SP in nasal secretions has been associated with the sensation of irritation due to exposure to organic solvents [27]. Although we found significant differences in the levels of SP between the groups, other environmental conditions such as UV light, wind, cold, and characteristics of outdoor work may also be involved in the increase of this neuropeptide in tears.

A central finding of the present study is that exposure to occupational levels of agrochemicals affects both low- and high-level visual functions. These results mirror our previous work examining the effect of organic solvent exposure on visual function. Significantly, individuals exposed to agrochemicals are associated with poorer detection (in some cases substantial) of basic visual features such as contrast, colour discrimination and the detection of global form and motion coherence. Visual function due to pesticide exposure may be via two potential mechanisms. First, chemical absorption via the ocular system and chemicals may lead to a direct change in the functioning of retinal tissue responsible for contrast and colour detection by retinal ganglion cells. Importantly, individuals in the exposed group have higher levels of SP in response to chemical exposure as a possible route. Second, through ingestion or via the respiratory system and accumulation in the body and potentially the brain. Indeed, recent evidence suggests that repeated exposure to pesticides may alter and cross the blood-brain barrier, leading to their accumulation in neural tissue and altering their function throughout the brain [28]. Note that high-level visual analysis occurs in numerous areas of the occipital and temporal lobes, which suggests that function change from agrochemical exposure is likely to be extensive. The diffuse effect of agrochemical exposure is supported by previous studies that report deficits in cognitive function and attention, which are mediated by mechanisms in the

prefrontal and frontal cortex [29]. The significance of these findings is that pesticide exposure can lead to poorer vision and visual perception, which has implications for visually guided behaviour and raises questions about the safety of prolonged use and exposure to agrochemicals.

In the present research, individuals comprising the exposed groups worked in the agricultural system focused on farming flowers. As mentioned, this group were exposed to a wide range of agrochemicals, typical amongst which were organic pesticides such as OPs and CAs. Given that workers were exposed to various chemicals, it remains entirely unclear whether one or a combination of chemicals causes neurotoxic symptoms and accounts for our finding of visual deficits. Previous studies have reported that exposure to organic solvents leads to similar deficits in visual function[12, 14, 21, 24]. Organic solvent exposure may cause a visual change in the present study, given that many chemicals/pesticides used in the exposed group were also organic solvents. In the present study, it was not possible to definitively determine which chemical/chemicals are the direct cause for the reported symptoms and visual function changes reported. Future studies are needed to investigate whether one or a combination of chemicals is responsible for neurotoxicity.

The measures of pesticide exposure (such as neurotoxicity such as the modified Q16, SP and AChE) were taken along with measurements of visual function. A potentially informative analysis is to determine whether these measures are correlated to establish whether there is an association between symptoms/levels of neurotoxicity and exposure and visual function change. However, such analyses have been conducted and have typically reported little or no relationship between subjective measures such as questionnaires and visual function changes [30]. This may be because the questionnaire symptoms score generally reflect attitudes and opinions, which are likely to be influenced by subjective and individual criteria for pain and comfort, which do not have to bear a relationship with functional measures. It is possible that once exposure has been reached to alter visual function, surpassing this level may lead to more general symptoms of neurotoxicity but not necessarily translate to greater deficits in function. Analogous to previous studies, this research finds no association between measures of neurotoxicity and exposure to agrochemicals and visual function.

This research reported an association between exposure to occupational levels of agrochemicals, including pesticides and alterations in both low- and high-level visual function. The exposed group reported higher neurotoxicity symptoms, and SP levels were elevated, suggesting an impact of chemical exposure on the body and brain. These individuals also showed poorer performance in detecting visual features such as contrast and colour and deficits in perception associated with higher-level visual processing, particularly in detecting global form and motion coherence. Significantly, our findings highlight the potential wide-ranging harmful effects of regular exposure to agrochemicals used in agricultural production.

It is important to note that it was not possible to ascertain whether deficits in neurological and visual function existed in the exposed group before participating in the present study. While care was taken to match both groups, and including a control group allowed a baseline point of comparison, future studies may wish to measure neurological and visual function in new workers, to definitively rule out any pre-existing conditions that may account for group differences.

## Supporting information

**S1 Data.**
(ZIP)

## Acknowledgments

This work was supported by Universidad de La Salle, Bogota, Colombia, and The University of New South Wales. We express our sincere gratitude to the flower farm workers from Cundinamarca, Colombia, especially those from the municipality of Sibate. We also acknowledge Juan Carlos Burgos from Gambur Flowers, Gerardo Andres Dussan, Mavel Rocio Cardenas, and Carlos Ivan Chaparro, who supported the data collection.

## Author Contributions

**Conceptualization:** Sieu K. Khuu.

**Formal analysis:** Ingrid Astrid Jiménez Barbosa, Sieu K. Khuu.

**Funding acquisition:** Ingrid Astrid Jiménez Barbosa, Martha Fabiola Rodríguez Alvarez, Laila Cristina Bernal Bechara.

**Investigation:** Martha Fabiola Rodríguez Alvarez, Laila Cristina Bernal Bechara.

**Methodology:** Ingrid Astrid Jiménez Barbosa, Martha Fabiola Rodríguez Alvarez, Laila Cristina Bernal Bechara, Sieu K. Khuu.

**Resources:** Martha Fabiola Rodríguez Alvarez.

**Software:** Sieu K. Khuu.

**Validation:** Ingrid Astrid Jiménez Barbosa.

**Writing – original draft:** Ingrid Astrid Jiménez Barbosa, Martha Fabiola Rodríguez Alvarez, Laila Cristina Bernal Bechara, Sieu K. Khuu.

**Writing – review & editing:** Ingrid Astrid Jiménez Barbosa, Martha Fabiola Rodríguez Alvarez, Laila Cristina Bernal Bechara, Sieu K. Khuu.

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
