## [Decision Letter · Decision Letter 0]

12 Apr 2023

PONE-D-23-05932Impairments in visual low- and high-level visual function associated with agrochemical usePLOS ONE

Dear Dr. Jiménez Barbosa,

Thank you for submitting your manuscript to PLOS ONE. After careful consideration, we feel that it has merit but does not fully meet PLOS ONE’s publication criteria as it currently stands. Therefore, we invite you to submit a revised version of the manuscript that addresses the points raised during the review process.

We look forward to receiving your revised manuscript.

Kind regards,

Samuel Adelani Babarinde, PhD

Academic Editor

PLOS ONE

Journal Requirements:

"We thank Gadiel Dumlao for his help with participant testing. This work was supported by Australian NHMRC grants APP1046198 and APP1085404 and an ARC discovery project DP140101560 awarded to JP and a Career Development Fellowship APP1049596 awarded to JP."

"This work was supported by Australian NHMRC (https://www.nhmrc.gov.au/) grants APP1046198 and APP1085404 and an ARC (https://www.arc.gov.au/) discovery project DP140101560 awarded to JP and a Career Development Fellowship APP1049596 awarded to JP."

Reviewers' comments:

Reviewer's Responses to Questions

**Comments to the Author**

1. Is the manuscript technically sound, and do the data support the conclusions?

Reviewer #1: Partly

Reviewer #2: Yes

2. Has the statistical analysis been performed appropriately and rigorously? 

Reviewer #1: Yes

Reviewer #2: Yes

3. Have the authors made all data underlying the findings in their manuscript fully available?

Reviewer #1: No

Reviewer #2: Yes

4. Is the manuscript presented in an intelligible fashion and written in standard English?

Reviewer #1: Yes

Reviewer #2: No

5. Review Comments to the Author

Reviewer #1: Comments

1. Title- The article is not only about the visual function of the participants but it is also assessing the neurotoxic symptoms so this should be captured in the topic to be something like ‘ Impairment of visual and neurologic functions associated with agrochemical use’

2. The presence of Substance P in the tear film actually shows that there is an ocular surface inflammation going on but this only could not specifically be a measure of ocular surface damage because the amount of it’s presence in individuals could be affected by other factors. A better measure of the ocular surface damage should be done using Slip lamp biomicroscope of the cornea in which epithelial defect can be identified or any other defect on any layer of the cornea OR you do an Anterior Segment Optical coherence tomography Or any other imaging modality to assess the ocular surface.

3. This study assessing the effect of organic solvents on ocular surface of workers should be a prospective cohort study and not a cross sectional study. Although controls were used but all these tests (visual and neurologic) were not done for the participants before stating the work on the agricultural farm. So how are you sure your findings were developed due to the exposure of the agrochemicals? Some might have had a mild premorbid ocular condition that does not warrant any complaints before so using verbal complaints are not enough. Subjective self assessment of the ocular status is not enough because it can be different in different individuals.

4. Are the two groups sex and age matched? Because some of the tests especially the contract sensitivity test varies with age. So if there is marked age difference then the comparision will be faulty.

5. The findings on the visual acuity was not reported at all in this article…. Is it similar in the two groups? Because poor visual acuity will definitely after colour vision and contrast sensitivity. And Visual acuity is the main measure of visual function, all other test are secondary. Was it affected at all in this study?

6. Who did the direct ophthalmoscopy and who interpreted the images from the digital retinograph system? These assessments need to be reviewed by an ophthalmologist before you can say that the findings are essentially normal. Because any of them might have Retinal dystrophy that is not yet symptomatic and such is better identified by an ophthalmologist.

7. Did the author provided spectacles for all those who have refractive errors free of charge? Or are they all on insurance that can ensure they get their spectacles without paying out of pocket because ‘cost’ can be barrier to uptake of spectacle use.

8. The authors did not mention the procedure and who took the biological samples(Blood and tears) Was necessary precaution taken while taking the samples to ensure samples where not contaminated because these can affect the results gotten.

9. How was the modified Q16 neurotoxic symptoms questionnaire administered? Is it self administered or interview administered? This is important to ensure that all the participant had a good understanding of the questions asked because the participants have different educational levels.

10. The authors did not mention the percentages of the workers that usually uses or don’t use their protective clothings while using these chemicals. This is very important because it directly determine the percentages of the workers directly exposed to these chemicals.

Reviewer #2: The topic needs to be re-framed to indicate that it is a controlled study eg the title may read: "IMPAIRMENTS IN HIGH AND LOW VISUAL FUNCTIONS ASSOCIATED WITH EXPOSURE TO AGROCHEMICALS: A CONTROL STUDY ". Suggestion has been included in the review. Some grammatical constructions also need to be readjusted appropriately.

6. PLOS authors have the option to publish the peer review history of their article (what does this mean?). If published, this will include your full peer review and any attached files.

Reviewer #1: No

Reviewer #2: **Yes: **Susannah Temitope Adepoju

---

## [Author Response · Author response to Decision Letter 0]

8 Jun 2023

Dear, Dr Samuel Adelani Babarinde

Thank you for handling the review of our manuscript. We thank you and the reviewers for their constructive comments. In the interest of providing a clearer submission, changes have been made to our manuscript in line with their suggestions. These changes are detailed point by point below. Hopefully with these changes a decision can be made regarding the suitability of our manuscript for publication in PloSONE.

Reviewer #1 comments: 

1. Title- The article is not only about the visual function of the participants but it is also assessing the neurotoxic symptoms so this should be captured in the topic to be something like ‘Impairment of visual and neurologic functions associated with agrochemical use.’

Answer: We agree that the title requires change to better reflect the aims and findings of the study. Both reviewers suggested appropriate titles, and in discussion with the authors we agree that the title suggested by reviewer 1, is a better description of the study. The title now has been changed. 

2. The presence of Substance P in the tear film actually shows that there is an ocular surface inflammation going on, but this only could not specifically be a measure of ocular surface damage because the amount of it’s presence in individuals could be affected by other factors. A better measure of the ocular surface damage should be done using Slip lamp biomicroscope of the cornea in which epithelial defect can be identified or any other defect on any layer of the cornea OR you do an Anterior Segment Optical coherence tomography or any other imaging modality to assess the ocular surface.

Answer: The subtitle in lines 157-158 has been changed to “Measures of neurotoxicity exposure and levels of substance P in tears”.

3. This study assessing the effect of organic solvents on ocular surface of workers should be a prospective cohort study and not a cross-sectional study. Although controls were used but all these tests (visual and neurologic) were not done for the participants before stating the work on the agricultural farm. So how are you sure your findings were developed due to the exposure of the agrochemicals? Some might have had a mild premorbid ocular condition that does not warrant any complaints before so using verbal complaints are not enough. Subjective self-assessment of the ocular status is not enough because it can be different in different individuals.

Answer: Note that in the present study, there was no “study period” and the participants were not measured at multiple time points. In our study, participants were measured only at once, and comparisons were made between the exposed and non-exposed groups at only one point. Therefore, it was not a prospective cohort study, but rather a cross-sectional case-control study. We raise the limitations of this design in the general discussion to alert readers to the possibility that symptoms and visual dysfunction may have existed prior to pesticide exposure (lines 485-490).

4. Are the two groups sex and age matched? Because some of the tests, especially the contract sensitivity test, varies with age. So, if there is marked age difference then the comparison will be faulty.

Answer: The average age of the exposed group was 35.2 years (SD±9.2 years), and 82.0% were women (line 289-290). In the control group, the average age was 30.69 years (SD±8.3 years), and 82.05% were women (line 305-306). According to the data, the two groups are matched.

5. The findings on the visual acuity was not reported at all in this article…. Is it similar in the two groups? Because poor visual acuity will definitely after colour vision and contrast sensitivity. And Visual acuity is the main measure of visual function, all other tests are secondary. Was it affected at all in this study?

Answer: In the study, all visual tests for both cases and controls were performed with optical correction. The worst best-corrected visual acuity accepted in all participants was 0.1 log MAR. This point is clarified in line 187-189.

6. Who did the direct ophthalmoscopy and who interpreted the images from the digital retinograph system? These assessments need to be reviewed by an ophthalmologist before you can say that the findings are essentially normal. Because any of them might have Retinal dystrophy that is not yet symptomatic, and such is better identified by an ophthalmologist.

Answer: Direct ophthalmoscopy and the images were evaluated by two optometrists’ researchers with PhD in Optometry and Master's in Vision Science, both under the guidance of a retinal ophthalmologist.

7. Did the author provide spectacles for all those who have refractive errors free of charge? Or are they all on insurance that can ensure they get their spectacles without paying out of pocket because ‘cost’ can be barrier to uptake of spectacle use.

Answer: All participants have health insurance that cover their spectacles. By law, companies are required to affiliate their workers with the health security system, which includes health insurance coverage.

8. The authors did not mention the procedure and who took the biological samples (Blood and tears) Was necessary precaution taken while taking the samples to ensure samples were not contaminated because these can affect the results obtained.

Answer: All samples were taken by a certified professional bacteriologist (line 171) who is trained and qualified to perform this type of sampling.

9. How was the modified Q16 neurotoxic symptoms questionnaire administered? Is it self-administered or interview administered? This is important to ensure that all the participants had a good understanding of the questions asked because the participants have different educational levels.

Answer: The Q16 neurotoxic symptoms questionnaire was administered through an interview process, with the interviewer recording the participant's response, and providing an opportunity for the participant to ask questions or clarify any doubts. It was not self-administered (as clarified in lines 164-167).

10. The authors did not mention the percentages of the workers that usually uses or don’t use their protective clothing’s while using these chemicals. This is very important because it directly determines the percentages of the workers directly exposed to these chemicals.

Answer: As mentioned in lines 130-133, although the use of protective equipment is mandatory by law, all workers reported not always using them. Unfortunately, a checklist was not conducted to confirm the use of protective equipment during each worker's work shift, and no percentages of usage of these elements were mentioned. This issue is now mentioned in the revised manuscript (see lines 130-133). 

Review #2 comments:

1. The topic may be modified to better convey the methodology; it may be rendered as “Impairment in high and low visual functions associated with agrochemical exposure: a controlled study “.

Answer: The title has been changed to “Impairment of visual and neurologic functions associated with agrochemical use” as suggested by reviewer 1 as we feel that this more appropriately describes the present study. 

2. There is need for references to be provided to support the assertion that several studies have been done to recognize the deleterious effect of pesticides on brain function in the introduction.

Answer: Additional references were added in line 72, and posteriorly, we mention more studies in line 83 and line 85.

3. Reference(s) need(s) to be provided to give more information on the Modified-16Q questionnaire and how it is scored in assessing neurotoxicity of pesticides. No search material matches the reference provided.

Answer: The links to the three articles are attached here and were added as references 14-16 in line 103 and line 168.

Reference 14: Jiménez-Barbosa IA, Boon MY, Khuu SK. Exposure to organic solvents used in dry cleaning reduces low and high level visual function. PLoS One. 2015;10: 1–23. doi:10.1371/journal.pone.0121422.

Available https://journals.plos.org/plosone/article?id=10.1371/journal.pone.0121422

Reference 15: Jiménez Barbosa ÍA, Khuu S y Ying Boon M. Modificación del cuestionario de síntomas neurotóxicos (Q16). Cienc Tecnol Salud Vis Ocul. 2011;(1): 19-37. https://ciencia.lasalle.edu.co/cgi/viewcontent.cgi?article=1029&context=svo

Reference 16: Jimenez- Barbosa Ingrid Astrid. Visual Function in Dry Cleaners. The University of New South Wales. 2014. https://doi.org/10.26190/unsworks/16756
http://hdl.handle.net/1959.4/53436. 

Available: http://www.unsworks.unsw.edu.au/primo_library/libweb/action/dlDisplay.do?vid=UNSWORKS&docId=unsworks_12131

4. To aid the reproducibility of this research, the distance of the place of residence of controls stated as “sufficiently away from agricultural farms” should be specified. So also, the estimated maximum number of times of exposure of the controls to the agrochemicals should be stated.

Answer: The controls resided at an approximate distance of 15 km from the farm where the workers were located (line 114-115). The controls were individuals who lived and worked in the city and had no direct contact with pesticides which the level of neurotoxicity was less than 32 in the modified Q16 questionnaire (i.e., less than low level) (line 111-112).

---

## [Decision Letter · Decision Letter 1]

4 Aug 2023

Impairment of visual and neurologic functions associated with agrochemical use

PONE-D-23-05932R1

Dear Dr. Barbosa,

We’re pleased to inform you that your manuscript has been judged scientifically suitable for publication and will be formally accepted for publication once it meets all outstanding technical requirements.

Kind regards,

Samuel Adelani Babarinde, PhD

Academic Editor

PLOS ONE

Additional Editor Comments (optional):

Reviewers' comments:

Reviewer's Responses to Questions

**Comments to the Author**

1. If the authors have adequately addressed your comments raised in a previous round of review and you feel that this manuscript is now acceptable for publication, you may indicate that here to bypass the “Comments to the Author” section, enter your conflict of interest statement in the “Confidential to Editor” section, and submit your "Accept" recommendation.

Reviewer #1: All comments have been addressed

Reviewer #2: All comments have been addressed

2. Is the manuscript technically sound, and do the data support the conclusions?

Reviewer #1: Yes

Reviewer #2: Yes

3. Has the statistical analysis been performed appropriately and rigorously? 

Reviewer #1: Yes

Reviewer #2: Yes

4. Have the authors made all data underlying the findings in their manuscript fully available?

Reviewer #1: Yes

Reviewer #2: Yes

5. Is the manuscript presented in an intelligible fashion and written in standard English?

Reviewer #1: Yes

Reviewer #2: Yes

6. Review Comments to the Author

Reviewer #1: The authors had made the necessary corrections and i am satisfied with their responses. Comments are satisfactory

Reviewer #2: (No Response)

7. PLOS authors have the option to publish the peer review history of their article (what does this mean?). If published, this will include your full peer review and any attached files.

Reviewer #1: **Yes:**

Reviewer #2: **Yes: **

---

## [Editor Report · Acceptance letter]

11 Aug 2023

PONE-D-23-05932R1 

Impairment of visual and neurologic functions associated with agrochemical use 

Dear Dr. Jiménez Barbosa:

I'm pleased to inform you that your manuscript has been deemed suitable for publication in PLOS ONE. Congratulations! Your manuscript is now with our production department. 

Kind regards, 

on behalf of

Dr. Samuel Adelani Babarinde 

Academic Editor

PLOS ONE